# Surging Glaciers in High Mountain Asia between 1986 and 2021

**Xiaojun Yao** [1] , **Sugang Zhou** [1,2,*] , **Meiping Sun** [1] , **Hongyu Duan** [1] and **Yuan Zhang** [1]

1   College of Geography and Environmental Science, Northwest Normal University, Lanzhou 730070, China; xj_yao@nwnu.edu.cn (X.Y.); sunmeiping1982@nwnu.edu.cn (M.S.); 2020120254@nwnu.edu.cn (H.D.); zhangyuan@nwnu.edu.cn (Y.Z.)

2   College of Urban and Environmental Sciences, Northwest University, Xi'an 710027, China

\*   Correspondence: zhousugang@stumail.nwu.edu.cn

**Abstract:** High Mountain Asia (HMA) is one of the concentrated areas of surging glaciers in the world. The dynamic movement of surging glaciers not only reshapes the periglacial landscape but also has the potential to directly or indirectly trigger catastrophic events. Therefore, it is crucial to understand the distribution patterns, periodicities, and occurrence mechanisms of surging glaciers. Based on Landsat TM/ETM+/OLI remote sensing images from 1986 to 2021, a total of 244 surging glaciers were identified in HMA in this study, covering an area of 11,724 km$^2$ and accounting for 12.01% of the total area of glaciers in this region. There are 185 surging glaciers identified within the Karakoram Range and Pamirs, which constitute the primary mountainous regions in HMA. From 1986 to 2021, these surging glaciers advanced at least 2802 times and exhibited different temporal and spatial patterns. A total of 36 glaciers in HMA experienced 2 or more surges during this period, with the highest number observed in the Pamirs (19), followed by the Karakorum (13), with the other regions having fewer occurrences. Obvious differences exist in the surge phase and the quiescent phase of glaciers in different regions of HMA. The surge phase of surging glaciers in the Karakoram Range and Pamirs is generally short, mostly in the range of 2~6 years. The quiescent phase lasts for 5~19 years and the overall surge cycle ranges from 9 to 24 years. The complex nature of glacier surges in HMA suggests that multiple mechanisms may be at play, necessitating further research.

**Keywords:** surging glacier; surge phase; quiescent phase; High Mountain Asia; Landsat

## 1. Introduction

An important part of the cryosphere, mountain glaciers are solid reservoirs of precious freshwater resources [1] and a sensitive indicator of climate change [2]. According to the Special Report on Global Warming of 1.5 °C released by the IPCC, High Mountain Asia (HMA) has been warming faster than the global average, which has caused the glaciers to shrink and thin rapidly in this region [3–5]. However, the glaciers in some regions of HMA exhibited different change trends [6]. For example, the glaciers in the Nyainqentanglha Mountains shrank rapidly, whereas the Karakoram Range, Pamirs, Pamir Alai region, and Kunlun Mountains witnessed relative stability, partial advancement, and even surging phenomena. These observations have garnered significant attention from the academic community [7–10].

A glacier surge is rapid movement of a glacier over a short period [11]. Surge-type glaciers oscillate between two stages of motion: the quiescent phase and the surge phase [12]. The surge phase, characterized by comparatively fast motion, occurs in quasiperiodic multiyear intervals [13]. During this phase, the sudden acceleration of the surging glacier leads to swift ice transfer without any change in total mass. This results in distinctive features such as medial moraine folds, ice surface fragmentation, sheared margins and bulging, overriding, and advancing fronts [11,14]. More significantly, the surging glacier poses a great threat to downstream infrastructure and residents' lives and property. Its rapid movement can destroy pastures, roads, bridges, villages, hydropower

stations, and other facilities in its path in a short period of time [15,16] and even trigger glacial lake outburst floods or block rivers to form dammed lakes [17–19].

Although the number of surging glaciers accounts for only 1% of total global glaciers, glacier surges have been observed in many regions, including Svalbard and East Greenland, Norway [20]; Yukon Territory, Canada; Alaska, U.S.A. [21]; and HMA [22–24]. HMA harbours the largest concentration of glaciers outside the polar regions [22], making it the most extensively developed cryospheric area in the middle and low latitudes. In recent years, the surging glaciers in HMA have attracted more attention, and some surging glaciers in the Karakoram Range [25–30], the Pamirs [31–35], West Kunlun [36–38], Tien Shan [39,40], and the Tanggula Mountains [41–43] have been identified. Due to the intricate terrain and climatic conditions in this region, coupled with the unpredictability of glacier surges, most of the literature on surging glaciers relies on satellite remote sensing data. The first inventory of surging glaciers in HMA (1861–2013) was completed by Sevestre and Benn [22], primarily based on a summary of the available literature and records. Recently, Vale et al. [23] identified a total of 137 surging glaciers in HMA between 1987 and 2019 utilizing the GEE platform and the GEEDiT tool. Guillet et al. [24] presented a regionally resolved inventory of surge-type glaciers based on their behaviour across HMA between 2000 and 2018, and identified 666 surging glaciers using a multifactor remote sensing approach that combined yearly ITS_LIVE velocity data, DEM differences, and very-high-resolution imagery (Bing Maps, Google Earth). Due to the differences in methodologies and time periods, the identification of surging glaciers has yielded inconsistent results, thereby limiting our understanding of their distribution patterns and occurrence mechanisms in HMA. The objectives of this study are as follows: (1) to identify surging glaciers in HMA between 1986 and 2021 from long time series of Landsat images; (2) to determine the occurrence time and frequency of these surging glaciers based on available yearly Landsat images; and (3) to investigate the periodicity and mechanisms of surging glaciers in HMA. It can provide a scientific basis and data support for understanding the characteristics of glacier changes in HMA and for glacier disaster prevention and mitigation efforts.

## 2. Study Area

High Mountain Asia (25°N~46°N, 67°E~103°E) mainly comprises the Qinghai–Tibet Plateau and surrounding alpine ranges, including the Himalayas, Hindu Kush, Karakoram Range, Pamirs, Tien Shan, Kunlun Mountains, Altun Shan, Qilian Shan, Nyainqentanglha Mountains, and Hengduan Shan (Figure 1), with an average altitude above 4000 m [44,45]. HMA functions as a water distribution system, termed the Asian Water Tower (AWT), that delivers water to almost two billion people [46]. Abundant glacial ice reservoirs and alpine lakes serve as significant sources of fresh water, while an extensive river system comprising the Yellow, Yangtze, Indus, Mekong, Salween, Ganges, Brahmaputra, Amu Darya, Syr Darya, and Tarim rivers provides fresh water to downstream regions [47,48].

According to the regional division scheme of the Randolph Glacier Inventory (RGI) V6.0 [49], HMA involves three regions: Central Asia (13), South Asia West (14), and South Asia East (15). It is home to a total of 95,536 glaciers covering an area of 97,605 km$^2$, accounting for 44.13% and 13.08% of global glacier coverage, respectively (Figure 1). Based on their temperature and surface velocity and the climatic conditions, the glaciers in HMA can be classified into three types: marine glacier, subcontinental glacier, and continental glacier [50]. Sevestre and Benn [22] estimated that there are a total of 2317 surging glaciers worldwide. The latest RGI V6.0 documented a count of 946 surging glaciers in HMA, including 337 possible surging glaciers, 208 probable surging glaciers, and 102 observed surging glaciers.

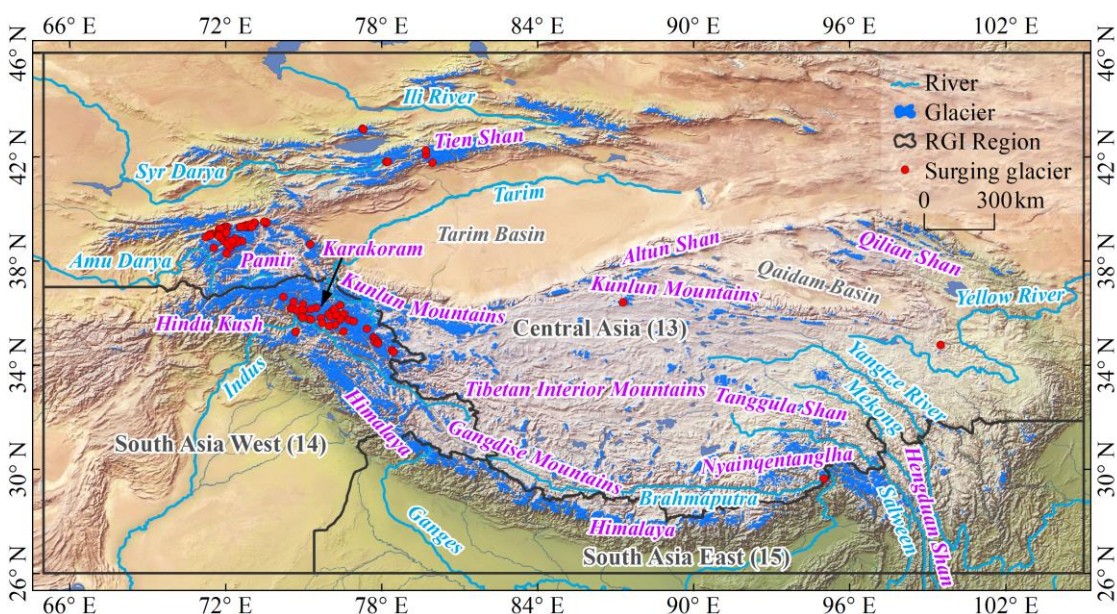

**Figure 1.** High Mountain Asia (HMA). The glacier outlines are obtained from RGI V6.0, which includes observations of surging glaciers (102). The background image is the Natural Earth II map with shaded relief, water, and drainages data from the Natural Earth website (https://www.naturalearthdata.com/, accessed on 22 March 2022).

## 3. Materials and Methods

### 3.1. Data

In this study, two authoritative glacier inventory datasets, namely, the Second Chinese Glacier Inventory (SCGI) and RGI V6.0, were adopted to revise glacier outlines during different periods. Among them, the SCGI data represents the status of glaciers in China except for Southeast Tibet during 2004–2011, and RGI V6.0 data serves as a fundamental reference for glaciers in other countries within the HMA region. They can be downloaded from the National Tibetan Plateau Data Center (https://data.tpdc.ac.cn/, accessed on 20 October 2021) and the GLIMS website (https://www.glims.org/, accessed on 2 December 2021), respectively.

The Landsat satellite series is renowned for providing the longest and most comprehensive archive of Earth observation data [51,52]. This extensive dataset offers substantial support for analysing long-term glacier changes [4,14,23,34,40,41]. The imagery utilized in this study consists of Collection 1 Level-1 Landsat 5 Thematic Mapper (TM)/Landsat 7 Enhanced Thematic Mapper Plus (ETM+)/Landsat 8 Operational Land Imager (OLI) data from 1986 to 2021, which were obtained from the United States Geological Survey (USGS) website (https://earthexplorer.usgs.gov, accessed on 25 January 2022). The acquired images were individually compared, and images with minimal cloud cover, little snow, and a clearly identifiable glacier terminus were selected to extract glacier outlines. A total of 7909 Landsat images were obtained, involving 112 Landsat satellite paths and rows (Figure 2a). The dataset comprises a total of 3419 Landsat TM images, 2417 Landsat ETM+ images, and 2073 Landsat OLI images. Notably, the number of available images remained relatively low until 1990; however, it consistently exceeded 200 scenes per year thereafter, with only a few exceptions in certain years (1991, 1992, 1995, 1997, and 2021) (Figure 2b). Furthermore, there is a higher abundance of imagery during the summer months (July to September), while other months exhibit comparatively fewer acquisitions (Figure 2c). All images underwent radiometric and geometric correction, as well as topographic correction based on DEM data. Subsequently, colour composites were generated by combining visible bands of Landsat TM/ETM+/OLI (Landsat TM and ETM+: bands 5, 4, 3; Landsat OLI: bands 6, 5, 2) using the CompositeBands tool in ArcGIS 10.4 software. Moreover, pan-sharpened images with

a spatial resolution of 15 m for Landsat ETM+/OLI were produced by fusing the colour composite with the panchromatic band using the CreatePansharpenedRasterDataset tool in ArcGIS 10.4 software. The above processes were batch-processed using Python code to enhance efficiency.

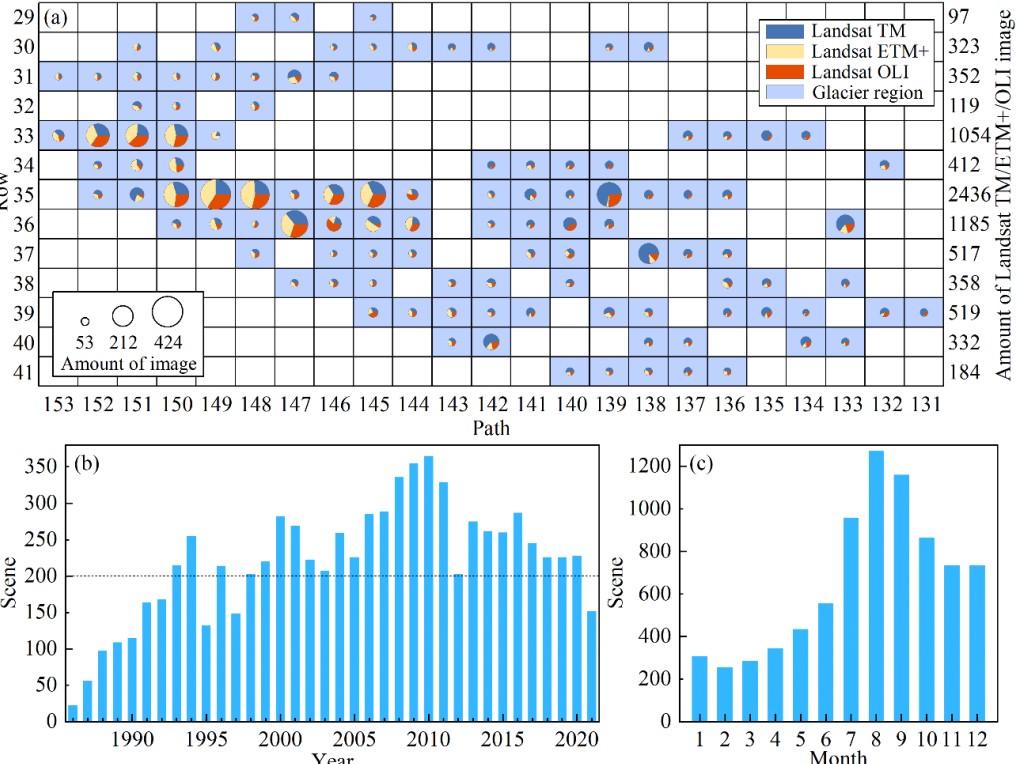

**Figure 2.** Landsat TM/ETM+/OLI images used to monitor surging glaciers in HMA. (**a**) The row and path refer to the orbit numbers traversed by the Landsat satellites. The pie chart illustrates the distribution of images captured by different sensors within the glacier region, while the right axis represents the cumulative number of each row. The size of each segment in the pie chart corresponds to the quantity of images acquired for each orbit number, with larger segments indicating a higher count and vice versa. (**b**,**c**) denote the annual and monthly image counts, respectively.

The rapid transfer of ice material during a glacier surge results in the redistribution of ice mass and obvious variations in surface elevation. Specifically, the reservoir zone of the glacier experiences thinning, while its receiving zone undergoes thickening [24]. Therefore, we utilized SRTM and ASTER DEM data to identify and validate the occurrence of partial glacier surging. The SRTM mission employed C-band radar from 11 to 22 February 2000, resulting in near-global data at a resolution of 30 m with a vertical accuracy of 16 m [53]. The ASTER DEM (AST14DEM) product is a digital elevation model generated by ASTER stereo pairs with a spatial resolution of 30 m [54]. The above DEM data were acquired from the NASA EARTH DATA website (https://earthdata.nasa.gov/, accessed on 13 March 2022). In the identification process of some surging glaciers, we obtained AST14DEM data for multiple periods ranging from 2000 to 2021.

The surface flow velocity of a surging glacier typically exhibits anomalies from the mean positive velocity, so we employed an analysis of glacier surface flow velocity to validate its surge. The ITS_LIVE (Inter-Mission Time Series of Land Ice Velocity and Elevation) data product is a part of the NASA 2017 MEaSUREs (Making Earth System Data Records for Use in Research Environments), which provides low-delay measurements of glacier and ice sheet surface velocity as well as elevation changes spanning 1985–2018 worldwide. These data were derived from Landsat TM/ETM+/OLI images and processed using the auto-Rift algorithm [7,55] with a spatial resolution of 240 m, downloaded from

NASA's Jet Propulsion Laboratory, California Institute of Technology website (https://its-live.jpl.nasa.gov/, accessed on 29 January 2022).

*3.2. Methods*

3.2.1. Glacier Outline Extraction

Various techniques have been employed to extract glacier boundaries from satellite remote sensing images [56–60]. In this study, we utilized a combination of automatic computer classification and visual interpretation, which is the most commonly used approach [40,61]. Our focus was on changes in glacier terminus positions, which were compared one by one using Landsat TM/ETM+/OLI images spanning 1986 to 2021 to identify significant advances. Moreover, glacier outlines were manually revised with the aid of the SCGI and RGI V6.0 datasets. The glacier outlines in the debris-covered zone were manually revised by considering image colour and texture features, distribution of glacial lakes, hydrological characteristics of glacial termini, and topographic attributes on both sides of each glacial and water system feature [61].

The accuracy of glacier outline extraction is mainly affected by remote sensing sensors, image registration [62], and pixel offset errors resulting from subjective visual interpretation [57,58,63]. The errors caused by satellite sensors and image registration are difficult to quantify [41,61]. Here, we just consider the error resulting from spatial resolution of remote sensing images, which can be calculated by the following equation [40,63]:

$$\varepsilon = N \times L \tag{1}$$

where $\varepsilon$ is area error (m$^2$); $N$ is the perimeter of a glacier outline (m); and $L$ is the side length of a single pixel (30 m, 15 m, and 15 m for Landsat TM/ETM+/OLI images, respectively).

3.2.2. Glacier Length Extraction

Glacier length is an essential attribute of glacier geometry, serving as the foundation for calculating the retreat and advance distance of a glacier terminus. This parameter effectively characterizes changes in glaciers and plays a crucial role in the global glacier inventory. Based on the automatic extraction method of glacier centrelines proposed by Zhang et al. [64], we extracted the length data of surging glaciers in HMA and took the maximum length as the statistical standard [65] to calculate the advance distance of surging glaciers.

The accuracy of glacier length extraction depends on the precision of the extracted glacier outline and the quality of DEM data. Generally, the influence of the latter on glacier length is negligible [64,65]. Therefore, the spatial resolution of a remote sensing image remains as the sole error source that can be calculated by the following equation [40]:

$$\rho = 1 - \frac{\lambda}{L} \tag{2}$$

where $\rho$ is the extraction accuracy of glacier length (%); $L$ is the glacier length (m); and $\lambda$ is the spatial resolution of the remote sensing image (30 m, 15 m, and 15 m for Landsat TM/ETM+/OLI images, respectively). The overall accuracy of the glacier length measurements in HMA reached 99.51%.

3.2.3. Glacier Surface Elevation Change Calculation

The penetration of snow ice by the SRTM C-band results in an underestimation of the surface elevation in glacier areas, whereas the penetration of snow ice by the SRTM X-band is negligible [66]. Consequently, we corrected the SRTM C-band data based on HMA's SRTM C/X-band penetration depth difference data presented by Jiang [67]. Prior to calculating changes in glacier surface elevation, it is essential to co-register different DEM data due to variations in their acquisition and processing methods. Therefore, we used the method proposed by Nuth and Kääb [68] to co-register the ASTER DEM and SRTM

DEM. Meanwhile, during the matching process, a threshold of $\pm$ 100 m was chosen to reject anomalies in elevation difference, and the area where the ground slope is less than 5° was excluded to improve matching accuracy [69]. The harsh environment in glacier regions makes it difficult to evaluate the error of DEM data using field measurements. Therefore, assuming that the elevation of non-glacier areas remains constant over time, the uncertainty of elevation change between multiple DEMs is assessed by calculating the mean elevation difference (MED) and standard deviation (SE) in non-glacier areas [70].

$$e = \sqrt{SE^2 + MED^2} \tag{3}$$

$$SE = \frac{STDV_{no\ glac}}{\sqrt{n}} \tag{4}$$

where $e$ is the error of elevation change; $STDV_{no\ glac}$ is the standard deviation of the nonglacial area; and $n$ is the number of pixels within the nonglacial area. Considering the strong spatial autocorrelation among neighbouring pixels in DEM data, it is possible to neglect such a correlation when the distance between pixels exceeds 20 times the spatial resolution of pixels [71]. Therefore, we adopted 600 m as the de-spatial autocorrelation distance.

### 3.2.4. Surging Glaciers Identification

Retreating, advancing, and surging glaciers often intermingle in the glacier zone, posing challenges to the accurate identification of target glaciers [72]. The most prominent feature of a surging glacier in remote sensing identification is its rapid terminus advance [73], followed by surface crevasses, ice folds, and moraine folds in the middle of the glacier [11,27,74]. Additionally, a substantial amount of material is transferred from the reservoir zone to the receiving zone during glacier surge and surface elevation changes [24,72]. Among the above processes, the rapid advance of a surging glacier terminus may result in a glacial lake outburst flood and infrastructure damage along its path. Therefore, our focus lies primarily on those glaciers with distinctive terminal changes. The glaciers exhibiting terminus advances were initially identified and marked based on the acquired images in this study. However, surging glaciers with a terminal advance of less than five pixels (TM) and ten pixels (ETM+ and OLI) were excluded due to their classification as advancing glaciers [40,75]. Consequently, this dataset may not include surface advancing glaciers where rapid terminus advance did not occur but mass flow was observed on the glacier surface (which would have limited downstream impact).

In this study, surging glaciers were identified based on the following criteria: (1) A glacier that experiences an increase in length of more than 150 m within one year is tentatively classified as a possible surging glacier. (2) Substantial and spatially concentrated changes in surface elevation, both in the reservoir area and at lower elevations (the receiving area), are considered indicative of mass redistribution typical of surging glaciers [24]. Therefore, we calculated changes in glacier surface elevation based on the available DEM data. If the surface elevation decreases in the reservoir area while increasing in the receiving area after a glacier surge, it further supports identification of the glacier as a surging glacier. (3) Surface characteristics and terminal shape of possible surging glaciers are observed through remote sensing images; moraine folds, cracks, and fragmentation on the glacier surface with a lobate or teardrop-shaped terminus serve as features for identifying such glaciers.

## 4. Results

### 4.1. Number and Distribution of Surging Glaciers in HMA

Based on the SCGI and RGI V6.0 datasets, in conjunction with Landsat TM/ETM+/OLI remote sensing images from 1986 to 2021, we identified a total of 244 surging glaciers in HMA by conducting comparative analyses of terminal positions, glacier surface features, glacier surface elevation changes, glacier flow velocity changes, and terminal shapes. These

glaciers have a total area of 11,724 km$^2$, accounting for 12.01% of the overall glacier area in HMA (Figure 3). It should be noted that among glaciers with main trunks and tributaries, some exhibited simultaneous surges in both the trunk and the tributary while others solely experienced surging in their tributaries. However, when expressing the area of these latter cases, it was calculated based on the entire extent of the respective glaciers; hence, this resulted in an overestimation of the total area occupied by surging glaciers.

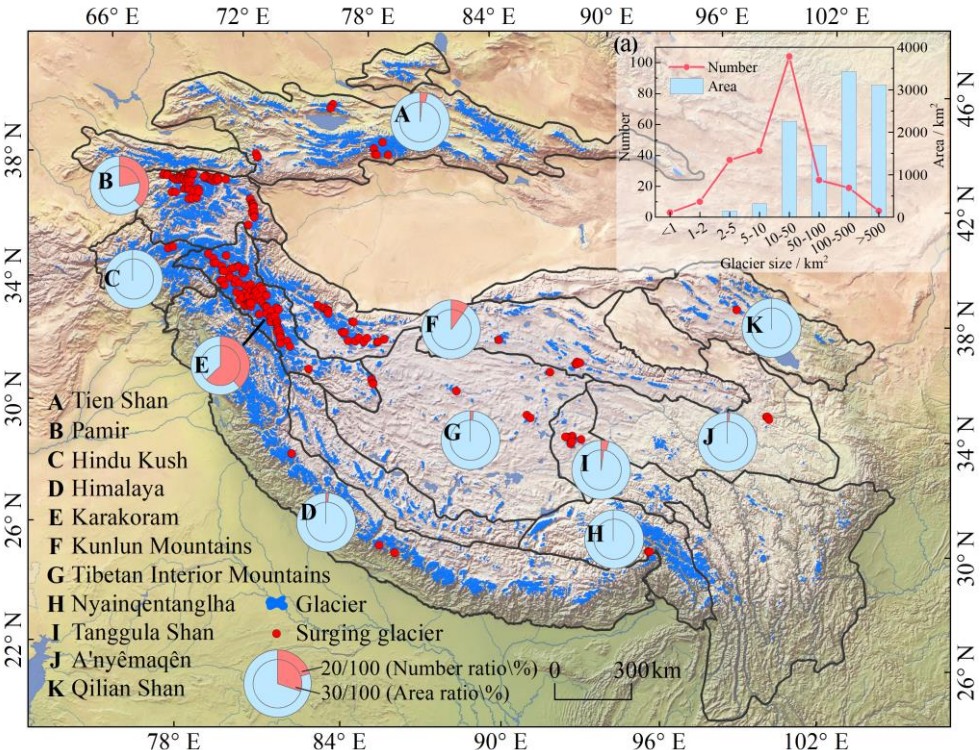

**Figure 3.** Distribution of surging glaciers in HMA. Figure (**a**) show the number and area of surging glaciers of different sizes.

According to the classification scheme of global mountain systems [76], the 244 surging glaciers are situated in the Karakoram, Pamir, Kunlun, Tien Shan, Tanggula, and Himalayan Mountains (Figure 3). Among them, a total of 185 surging glaciers with an area of 9934 km$^2$ are spatially distributed in clusters in the Karakoram Range and Pamirs, accounting for 75.82% and 84.71% of the total number and area of surging glaciers in HMA, respectively. In the Kunlun Mountains, there are 19 surging glaciers in West Kunlun and 5 in East Kunlun. The Qilian Shan only has one surging glacier, while Hengduan Shan and Altun Shan have none. In terms of area size (Figure 3a), the largest number of surging glaciers is observed in the range of 10–50 km$^2$ (104 glaciers), accounting for 42.62% of the total number, followed by those with an area of 5–10 km$^2$ (43 glaciers, 17.62%). Only a small proportion consists of glaciers with an area less than 2 km$^2$ (13 glaciers, 5.33%). Furthermore, the majority of surging glaciers in HMA are medium- to large-scale, as indicated by the fact that the total area covered by surging glaciers larger than 100 km$^2$ amounts to 7215 km$^2$ and accounts for 61.54% of the overall glacier area in HMA, while those ranging from 10 to 100 km$^2$ cover an area of approximately 4050.44 km$^2$ (34.55%).

## 4.2. Frequency of Glacier Surges in HMA

The criterion for identifying a glacier's advancement is the displacement of its terminus in the two images. We thoroughly examined all available remote sensing images to determine the frequency of glacier advances. The results reveal a total of 2802 advances from 244 surging glaciers in HMA from 1986 to 2021 (Figure 4). Surging glaciers exhibited advances every year throughout the study period. Prior to 1993, surges of glaciers occurred

less than 40 times per year in HMA. However, there was a rapid increase from 1993 to 2016, with advances exceeding 80 times per year except in limited images available on the USGS website in 1995 (40 times); this was followed by a gradual decrease after 2016. Among these years, surging glaciers advanced most frequently in 2010, 2011, 2004, and 2016 with occurrences of 116, 113, 113, and 112, respectively. This was followed by occurrences of 109, 108, 107, 106, and 105 in 2014, 1994, 2006, 2008, and 2007, respectively. The fewest instances of advance were observed in 1988, with only four occurrences.

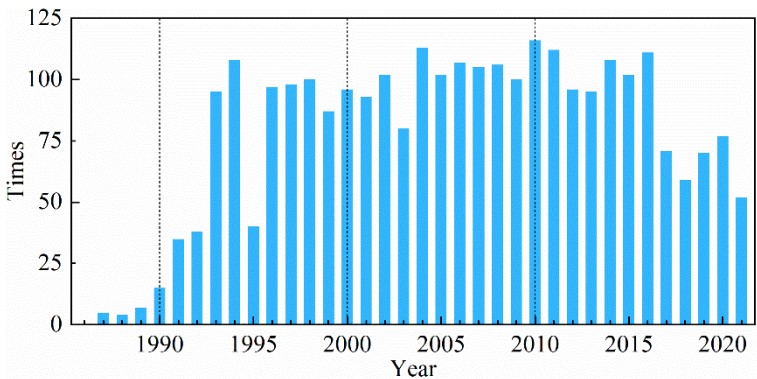

**Figure 4.** Frequency distribution of advances of surging glaciers in different years in HMA.

Among different mountain regions, the highest number of surging glacier advances occurred in the Karakoram Mountains (1401), followed by the Pamirs (902) and Kunlun Mountains (261). Fewer advances took place in other regions, with all below 100 (Figure 5). Annual occurrences of glacier surges were observed in both the Pamirs and Karakoram Mountains, with a rapid increase in the former from 1993 to 2017, peaking at over 20 times per year before gradually declining thereafter. Surging events in the latter have been on an upward trend since 1988, surpassing an annual count of 30 after 1993 except in 1995. The Kunlun Mountains saw relatively few surges until 2007, which began increasing annually before reaching their peak in 2015 and slowing down afterwards. No glacier surging events occurred in Tien Shan and Tanggula Shan after 2010 and 2015, respectively. Overall, these five mountains had a high incidence of surging glaciers between 1995 and 2010 but have seen a decreasing trend during the last 5–10 years.

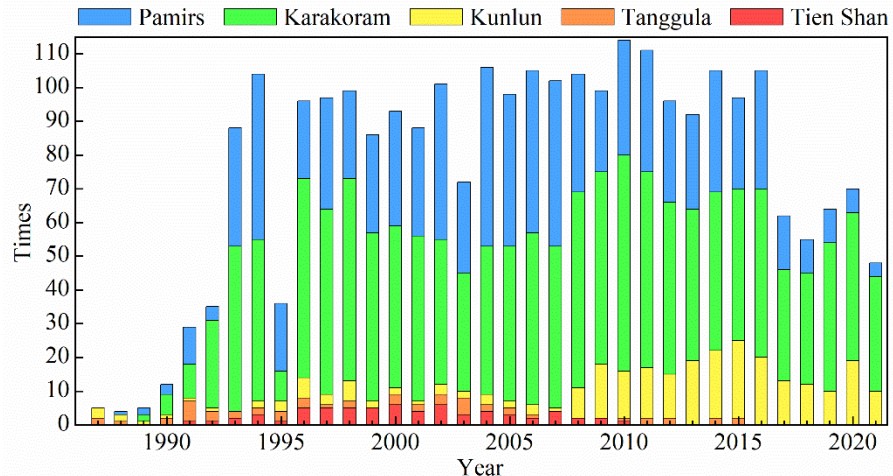

**Figure 5.** Distribution of the frequency of glacier advances in the main regions of HMA from 1986 to 2021.

*4.3. Occurrence Characteristics of Representative Surging Glaciers in HMA*

To clarify the occurrence characteristics of surging glaciers in HMA, the Oshanina Glacier (G071487E39039N), with the greatest terminal advance, and the Musta Glacier

(G076298E35850N), with the most tributaries, were selected as target glaciers. And we analysed the terminal changes in the surge phase as well as the changes in surface elevation and flow velocity before and after their surges.

### 4.3.1. Oshanina Glacier

The Oshanina Glacier, situated in the West Pamir region, had a length of 14.6 km and an area of 21 km$^2$ in 2000. Between October 2010 and August 2011, it advanced rapidly by 8286 m (30 m/d), resulting in an area increase of 4.9 ± 0.48 km$^2$ and causing it to have the longest advance distance in HMA during the study period (Figure 6). From October to December 2010, the Oshanina Glacier experienced an advance of 3950 m (62 m/d) with an area increase of 2.9 ± 0.23 km$^2$, reaching the peak of the surge. Subsequently, from January to August 2011, there was a terminus advance of 4336 m (18 m/d) and an area expansion of 1.96 ± 0.25 km$^2$. Notably, from June to July 2011, the terminus rapidly advanced by 2403 m (50.08 m/d), leading to an area increase of 0.85 ± 0.14 km$^2$ and marking the subpeak of the surge (Figure 7). The Landsat TM/ETM+ images, which include identifiable terminal positions of the Oshanina Glacier, were acquired in all months except March 2011. There were two or three available images for each individual month, providing a reliable basis for calculating the velocities of terminal advance during the specific month. For example, between 1st and 9th December in 2010, the Oshanina Glacier terminus advanced by 431 m at a speed of 54 m/d. Similarly, from 5 to 21 July in 2011, this glacier's terminus once again advanced by 826 m with a comparable speed (53 m/d).

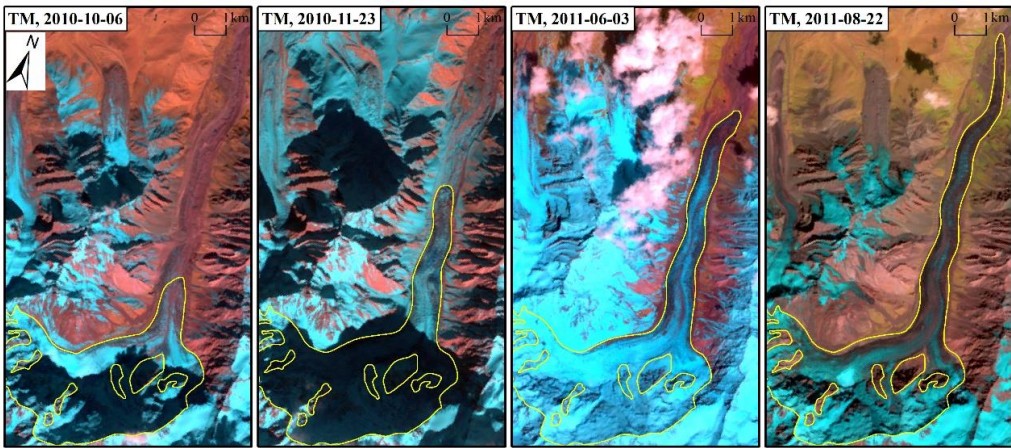

**Figure 6.** Changes in the Oshanina Glacier in the West Pamir from 2010 to 2011.

Based on the AST14DEM data acquired in 2009 and 2011, calculations were made to determine changes in glacier surface elevation before and after surging of the Oshanina Glacier. The results indicate a clear thinning in the reservoir area and thickening in the receiving area (Figure 8). Specifically, the reservoir zone experienced an average thinning of 68 ± 0.88 m with a maximum thinning of 157 ± 0.88 m, whereas the receiving zone exhibited an average thickening of 80 ± 0.88 m with a maximum thickening of 168 ± 0.88 m. It is evident that during the surging process, excess ice was transferred from the reservoir zone to the receiving zone, which is consistent with the observed advance at the glacier terminus.

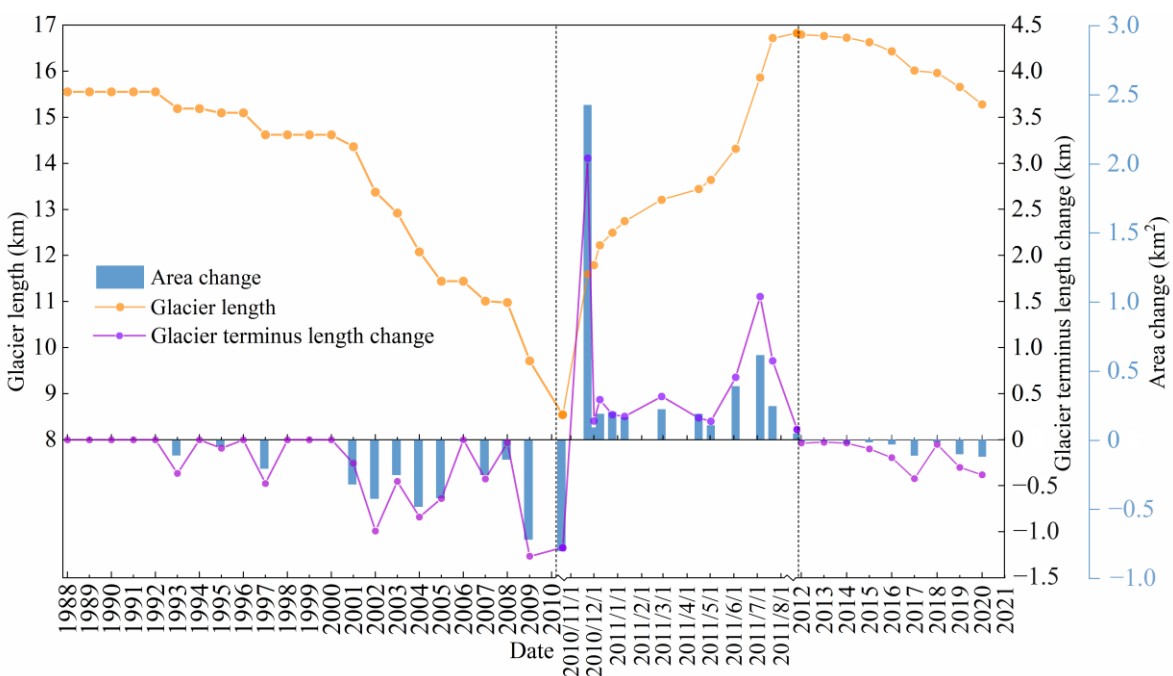

**Figure 7.** Variations in length, terminal advance distance, and area of the Oshanina Glacier in West Pamir during the surging phase.

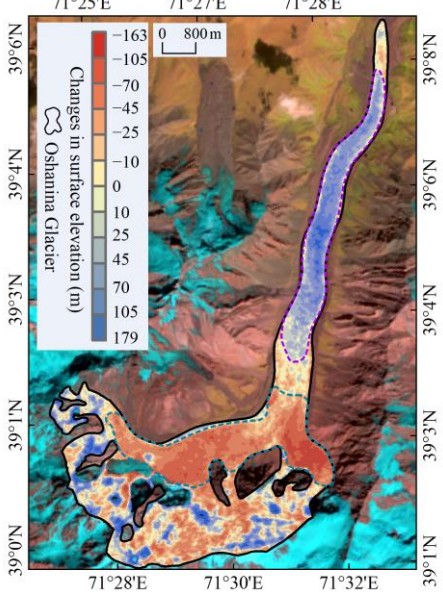

**Figure 8.** Changes in surface elevation of the Oshanina Glacier in the West Pamir from 2009 to 2011. Dotted lines in the figure: delft blue denotes the reservoir area and purple denotes the receiving area.

### 4.3.2. Musta Glacier

The Musta Glacier, located in the middle reaches of the Kelechin River Basin in the Karakoram Mountains, is a representative valley glacier encompassing numerous tributary glaciers and covering an area of 199 km². During the period of 1986–2021, several tributary glaciers (A1, A2, A3, A4) of the Musta Glacier underwent different degrees of surging; however, there was no significant change observed at the terminus of the trunk glacier (Figure 9). Among these tributaries, tributary glacier A1 commenced advancing in 1992 and ended in 2000, going forward a total of 2587 m (323 m/a) and increasing its area by 1.44 ± 0.21 km². The most significant advance occurred between 1996 and 1998, when glacier length and area increased by

1979 m (989 m/a) and $0.8 \pm 0.2$ km$^2$, respectively, reaching the surging peak. The terminus of this tributary merged into and overlaid the trunk glacier, with a lobed terminus and a distinctive pushback curl [77]. Tributary glacier A2 started surging the earliest (1991), had the longest surge phase (21 years), and advanced slowly from 1991 to 2004. Then, it advanced rapidly by 557 m and increased its area by $0.27 \pm 0.11$ km$^2$ from 2004 to 2005 prior to entering a deceleration phase. From 1991 to 2012, this glacier advanced by 2002 m, with an increased area of $0.85 \pm 0.1$ km$^2$. Tributary glacier A3 moved with the trunk glacier after merging with it, resulting in a squeezed and curved terminus. Compared to the other tributary glaciers, tributary glacier A3 moved slowly, advancing 1117 m with an increased area of $0.32 \pm 0.18$ km$^2$ from 1992 to 2000. Tributary glacier A4 began to surge in 2020, moving forward 1033 m (8.5 m/d), 589.4 m (24.5 m/d), and 737 m (23 m/d) from April to August, August to September, and September to October, respectively. Subsequently, the forward distance and movement speed slowed down to 354 m (10.9 m/d) from October to November and 702 m from January to November 2021. From 2020 to 2021, tributary glacier A4 moved forward a total of 3744 m and increased its area by $3.71 \pm 0.13$ km$^2$, and its terminus converged and overlapped with the trunk glacier, with a broken and fissured surface.

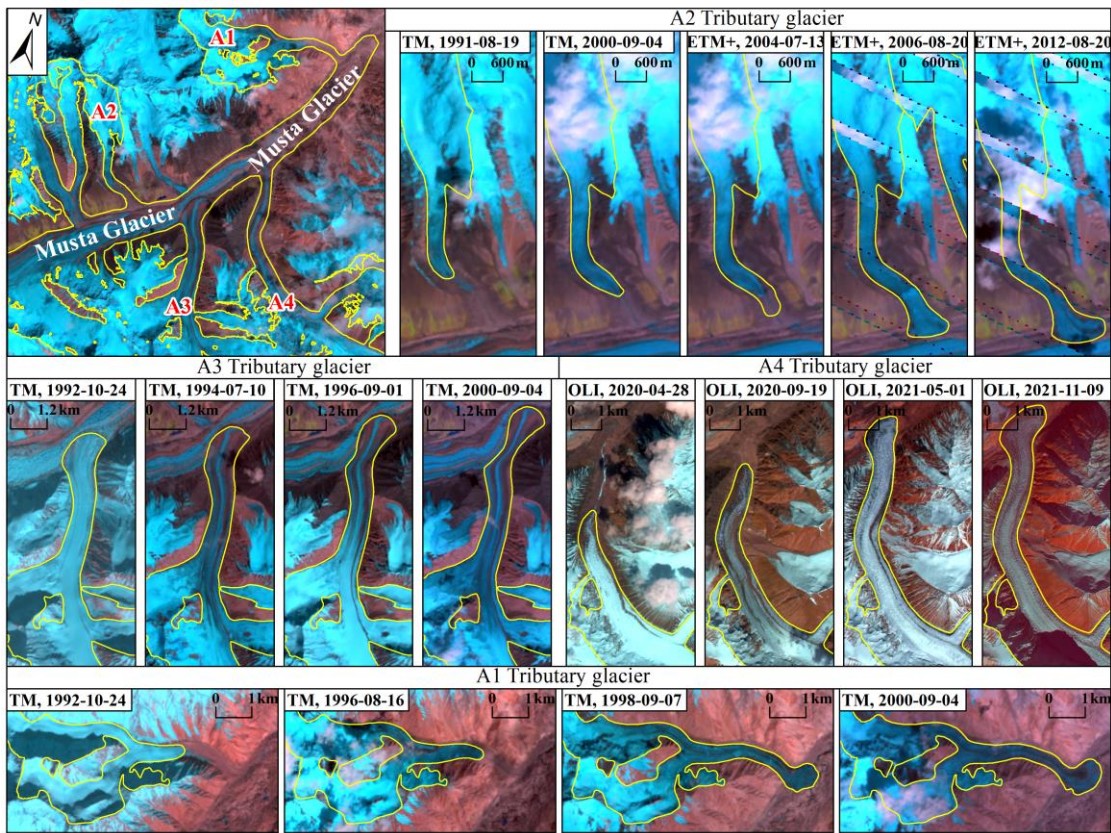

**Figure 9.** Surging process of the Musta Glacier.

Due to the limitations of available data, only changes in surface elevation of tributary glaciers A2 and A4 were calculated. The results show that the reservoir and receiving zones of the two glaciers had different degrees of thinning and thickening after the surge (Figure 10). From 2000 to 2014, the average thinning in the reservoir zone of tributary glacier A2 was $30 \pm 0.85$ m with a maximum value of $76 \pm 0.85$ m, while its receiving area experienced an average thickening of $20 \pm 0.85$ m with a maximum value of $49 \pm 0.85$ m. Tributary glacier A4 exhibited an average and maximum thinning of $65 \pm 1.33$ m and $214 \pm 1.33$ m in the reservoir area from 2019 to 2021 as well as an average and maximum thickening of $124 \pm 1.33$ m and $253 \pm 1.33$ m in the receiving area, respectively. It is evident

that tributary glacier A4 discharged excessive ice from the reservoir zone to the receiving zone during the active phase.

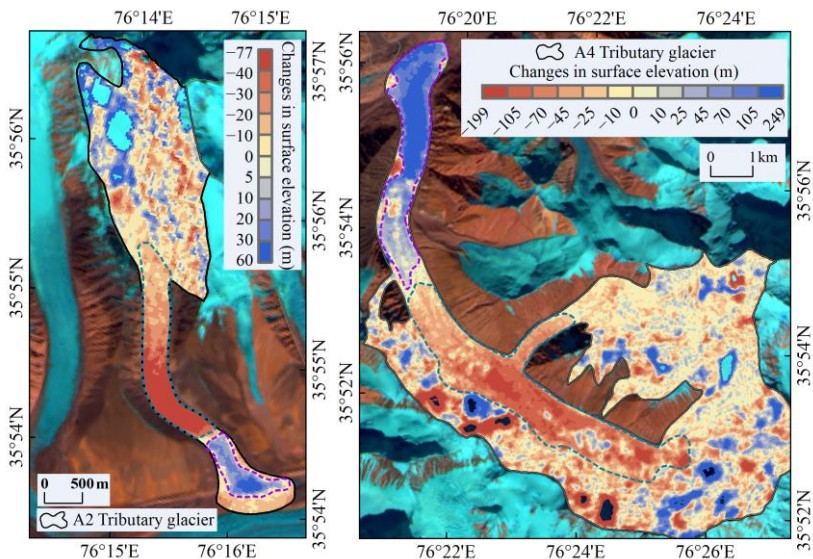

**Figure 10.** Surface elevation change in the Musta Glacier. Dotted lines in the figure: delft blue denotes the reservoir area and purple denotes the receiving area.

The ITS_LIVE glacier velocity data from 1986 to 2018 also revealed the relatively slow velocities of tributary glaciers A1 and A2 of the Musta Glacier (Figure 11). The velocity of tributary glacier A1 reached a maximum of 29.17 m/a in 2003–2005, while the terminus velocity of tributary glacier A2 has increased since 2006. Tributary glacier A3 had a faster flow velocity than tributary glaciers A1 and A2, with a higher flow velocity at 2.4 km from the terminus in 1989–2000, reaching a maximum velocity of 124 m/a. After 2011, there were higher flow velocities at 2.4–6 km from the terminus, and an increasing trend was observed. The flow velocity of tributary glacier A4 tended to increase year by year between 4.8 and 7.6 km from the terminus from 1986 and reached the maximum flow velocity (119 m/a) in 2018.

### 4.4. Glacier Surge Phase, Quiescent Phase, and Surge Cycle in HMA

The surge cycle is defined as the duration between two surges of a glacier, including the surge and quiescent phases [11,78]. The surge phase is the entire period from the beginning to the end of the glacier surge, while the quiescent phase is the interval between the end of one surge and the start of the next surge [11]. A total of 36 glaciers in HMA experienced 2 or more surges from 1986 to 2021 (Figure 12), with the most in the Pamirs (19), followed by the Karakoram (13) and Kunlun Mountains (2); the Himalayas and A'nyêmaqên Mountains had only 1 each.

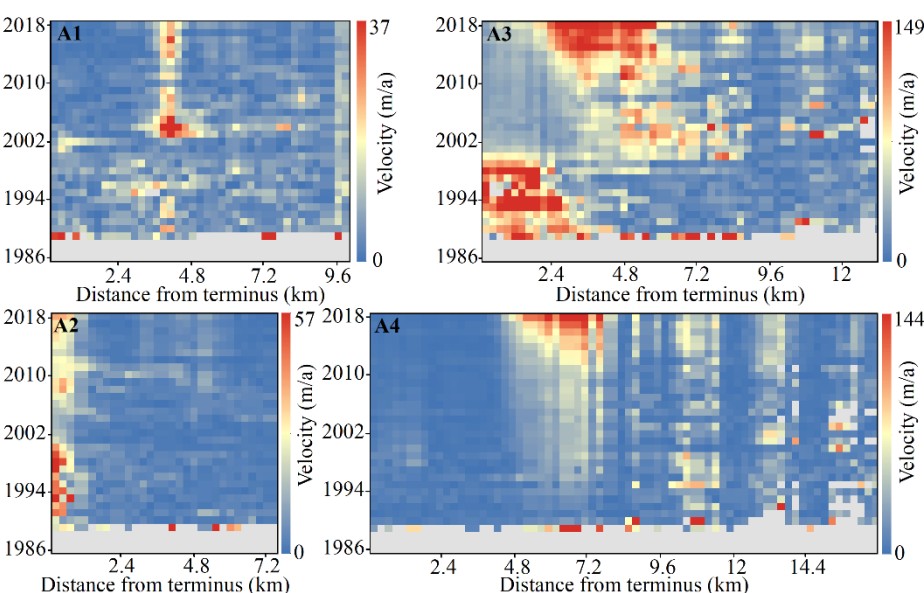

**Figure 11.** Mean annual velocity changes in the centreline of the Musta Glacier (tributary glaciers **A1–A4**).

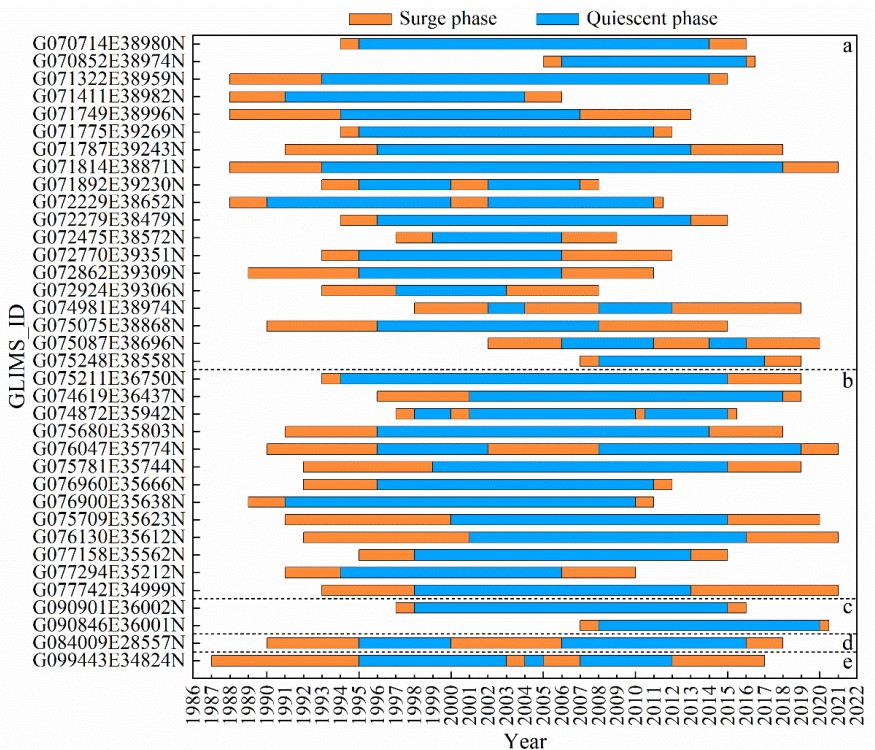

**Figure 12.** The surge phase, quiescent phase, and time of surge in different regions of HMA from 1986 to 2021. (The orange represents the surge phase, and the blue represents the quiescent phase; (**a**): Pamirs (**b**): Karakoram (**c**): Kunlun Mountains (**d**): Himalayas (**e**): A'nyêmaqên Mountain.)

    The surge phase of glaciers in the Pamirs lasts from several months to 7 years, while individual glaciers have longer durations, with quiescent phase ranging from 2 to 25 years. The surge cycle ranges from 5 years to 30 years, with the majority falling between 9 and 22 years. The surge phase of glaciers in the Karakoram region ranges from several months to nine years, with longer duration for individual glaciers. The quiescent phase ranges from 2 to 21 years, with the most glaciers (five) having a quiescent phase of 15 years, and the surge cycle ranges

from 3 to 24 years, with most cycles ranging from 22 to 24 years. The surging glaciers in the Kunlun Mountains have surge phases ranging from 1 to 24 years, and only two glaciers had a quiescent phase (12 and 17 years) and surge cycle (13 and 18 years). The surge phase of glaciers in the Himalayas is 2 to 6 years, the quiescent phase is 5 to 7 years, and the surge cycle is 10 to 13 years. The surge phase of glaciers in the A'nyêmaqên Mountain is one to eight years; the Qushi'an No. 17 Glacier (G099443E34824N) has potential ice avalanches, with a quiescent phase of eight years and a surge cycle of nine years. The duration of the surge phase varies from 1 to 19 years in the Tien Shan, 2 to 7 years in the Hindu Kush Mountains, 2 to 8 years in the inner area of the Qinghai–Tibet Plateau, and 3 years in the Qilian Shan.

## 5. Discussion

### 5.1. Similarities and Differences among Existing Surging Glacier Datasets in HMA

There exist three complete inventories [22–24] and multiple regional inventories [29,32,35,36,39,79,80] of surging glaciers in HMA. However, the number of surging glaciers in HMA varies widely (Table 1) due to differences in definition criteria, identification methods, study periods, and regions.

**Table 1.** Comparison of the number of surge-type glaciers in HMA identified by other studies and our results.

| Area | Time Range | Number | This Study | Data Source | Evidence [1] | Reference |
|------|-----------|--------|-----------|-------------|--------------|-----------|
| HMA | 1861~2013 | 946 | 244 | Literature | various | RGI V6.0 |
| | 2000~2018 | 666 | | DEM, ITS_LIVE, Google Earth, and Bing Maps | dh, dv, sf | [24] |
| | 1987~2019 | 137 | | Landsat | dt | [23] |
| Pamir | 1972~2006 | 215 | 91 | Resurs-F satellites, Landsat, Aster | dh, dv, sf | [32] |
| | 1988~2018 | 186 | | Landsat, Corona and Hexagon, AW3D30 DEM, SRTM DEM, Aster GDEM | dt, dh | [35] |
| Karakorum | 1960~2011 | 90 | 94 | Landsat, Aster | sf | [79] |
| | 1976~2012 | 101 | | Landsat, SAR satellite imagery | dt, dv | [80] |
| | 1840s~2017 | 221 | | Landsat and Aster | dt, dv, sf | [29] |
| West Kunlun | 1972~2014 | 31 | 24 | Landsat, Satellite Synthetic Aperture Radar Images | dt, dv | [36] |
| Tian Shan | 1964~2014 | 39 | 10 | Landsat, Corona KH-4, Hexagon KH-9, Cartosat, SPOT | dt, dh | [39] |

[1] dh, change in glacier surface elevation; dt, changes in glacier terminus position; dv, changes in glacier surface velocity; sf, glacier surface features.

Sevestre and Benn [22] were the first to conduct a comprehensive global inventory of surging glaciers, which was documented in the RGI data. Within the HMA region in RGI V6.0, there are 102 confirmed surging glaciers and 545 possible surging glaciers. In this study, we identified 57 of the 102 confirmed surging glaciers. However, we could not identify 45 surging glaciers in the RGI data due to their unclear advance during the study period or relatively short distance from the glacier terminus. Additionally, we identified 46 of the 545 possible surging glaciers. The remaining 475 unidentified surging glaciers are located in the Pamir with a relatively small average area of 3.7 km$^2$, of which 82.5% of the total glaciers are less than 5 km$^2$ and 27.2% are less than 1 km$^2$. Further comparison revealed that most glaciers remained unchanged or even retreated at their terminus during our study period, except for some advancing on the surface or terminus of individual glaciers, which is consistent with Guillet et al.'s findings [24]. Furthermore, it is worth noting that RGI V6.0 did not include the surging glaciers in the Kunlun Mountains and the Tanggula Shan or the Northern Inylchek and Samilowich Glaciers in the Tien Shan that were identified as surging glaciers in some studies [39,40].

Through comprehensive analysis of glacier surface elevation change, velocity change, and surface characteristics from 2000 to 2018, Guillet et al. [24] identified a total of 666 surging glaciers in HMA, which is significantly higher than our results (244). A comparison

shows that 222 surging glaciers were identified in common. The remaining 443 surging glaciers not identified in our study had no or only minor terminus advance. The method used by Guillet et al. [24] identified many surface advancing glaciers (i.e., glaciers that are not significantly advancing at their terminus but are advancing in their upper part) as surging glaciers [40], which were not identified in this study. On the other hand, we identified 21 distinct surging glaciers with clear terminal advances that were not detected in Guillet et al.'s study [24]. Some examples are shown in Figure 13 (glaciers coded G071749E39229N, G071926E39225N, and G075185E38699N, respectively).

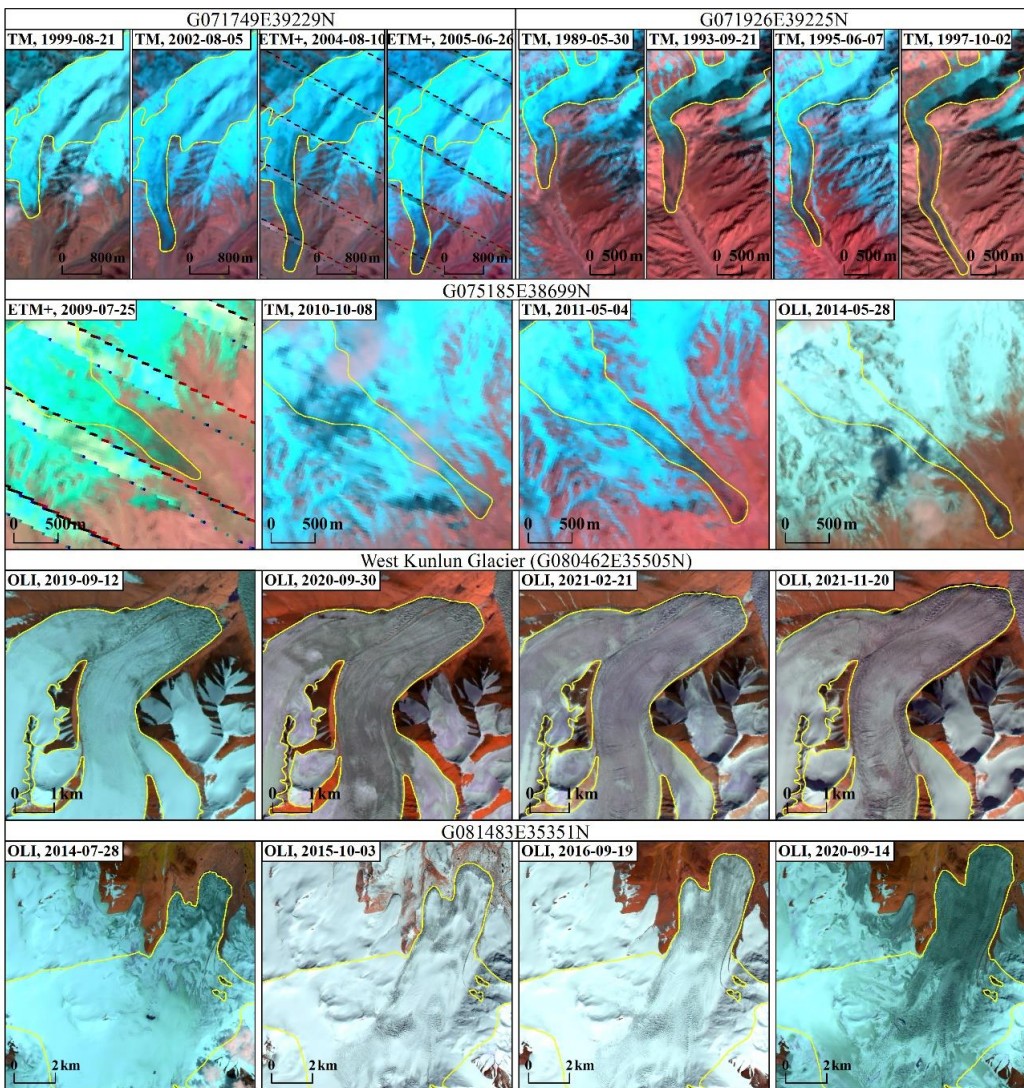

**Figure 13.** Examples of surging glaciers identified in this study but not included in other existing datasets.

Vale et al. [23] utilized GEEDiT, a tool developed by the cloud-based geospatial data platform GEE, to identify 137 surging glaciers from 1987 to 2019 in HMA. These glaciers are mainly located in the Karakorum, Pamir, West Kunlun, Tian Shan, and Himalayas. No surging glaciers were identified in other regions. In comparison with our results, there are 83 surging glaciers co-identified. The remaining 56 glaciers had shorter terminal advancement distances (less than 150 m), which were identified as advancing glaciers rather than surging glaciers in this study. Furthermore, we discovered an additional set of 161 surging glaciers with clear evidence of surge activity (Figure 13, glaciers coded G071749E39229N, G071926E39225N, and G075185E38699N, respectively).

Scholars have also investigated surging glaciers in some mountains or local regions in HMA. For example, Kotlyakov et al. [32] identified a total of 215 surging glaciers in the Pamirs from 1972 to 2006, while Goerlich et al. [35] found 206 surging glaciers from the 1960s to 2018, both using the same method as this study. Due to the exclusion of glaciers with small terminus advance distances by this study, the identified surging glaciers are fewer than those in the above two datasets, with the co-identified surging glaciers numbering 82. Copland et al. [79] identified 90 surging glaciers in the Karakorum Mountains from 1960 to 2011, but this result may be incomplete as it relied on glacier surface features (e.g., looped/folded medial, surface foliation, etc.). Rankl et al. [80] discovered an additional 10 surging glaciers based on Copland et al.'s result [79]. Bhambri et al. [29] identified 221 surging glaciers from the 1940s to 2017 but failed to capture some short-term surging glaciers due to limitations of early available satellite images. In contrast, our study provides more detailed evidence of short-term surging glaciers. Yasuda and Furuya [36] identified nine surging glaciers in West Kunlun, including four that surged from 1972 to 1992, and five that surged from 1992 to 2014. A comparison with our result shows that there are six co-identified surging glaciers. Three surging glaciers were not identified because their surging time is beyond the period of this study. In addition, we discovered two recently surging glaciers, namely, the West Kunlun Glacier (G080462E35505N) and the glacier coded G081483E35351N, respectively (Figure 13). The latter was identified as a possible surging glacier by Yasuda and Furuya [36]. The study conducted by Mukherjee et al. [39] and our own research identified 39 and 10 glaciers, respectively, in the Tien Shan region as surging glaciers. This discrepancy can be attributed to the criteria employed by these two studies; specifically, Mukherjee et al. [39] classified slowly advancing glaciers at their terminus as surging glaciers, whereas we categorized them as advancing glaciers. Overall, this study identified a relatively lower number of surging glaciers in HMA but provides valuable glacier boundary and length data for each surge event, serving as direct evidence of glacier surges and potentially offering a reference for future detailed investigations.

### 5.2. Periodicity of Surging Glaciers in HMA

The surge cycle of glaciers exhibits significant variations in different regions, while the timing of the surges and quiescent phases of the same glacier remains relatively constant [81]. Typically spanning years to several decades, glacier surge cycles can even extend over longer periods for individual glaciers [75]. Distinctions exist in terms of surge phase, quiescent phase, and surge cycle across various regions in HMA, with no exact replication of surge characteristics observed within the same region.

Previously, both data and studies on glacier surge cycles were scarce, with only a few glaciers described to have undergone a single surge cycle. For example, the surge cycle of Medvezhiy Glacier (G072229E38652N) in the Pamir spans approximately 10–14 years, while that of Bivachny Glacier (Tributary of glacier coded G072234E38780N) ranges from 15 to 20 years [73,74], and the surge cycle of Oshanina Glacier (G071487E39039N) is around 25 years [32,35]. In the East Pamir, the glacier coded G074348E39282N had a surge phase of about three years and a surge cycle of ten years [34], and the glacier coded 5Y663L0023 (RGI v6.0 code: G075022E38910N) had a surge phase of four years and a minimum quiescent phase of about fifteen years [73]. Our findings indicate that the surge cycle of two glaciers in the Kunlun Mountains were 13 and 18 years, respectively. However, previous studies showed that glaciers in the West Kunlun region exhibit longer surge phases exceeding 5 years along with surge cycles surpassing 42 years [36]. Furthermore, Guo et al. [82] speculated that the surge cycle of the Yulinchuan Glacier (G087301E36380N) is at least greater than 40 years, which differed from our results. The duration of the surge phase in Tien Shan varied from 1 to 19 years, accompanied by quiescent phases ranging from 30 to 50 or more years and overall surge cycles roughly equal to or exceeding 50 years [40]. The study period of this paper (1986~2021) benefited from a greater availability of remote sensing images, enabling improved monitoring of glacier surging occurrences as well as identification of additional surge cycles and quiescent phases.

There are differences in glacier surge phase, quiescent phase, and surge cycle in other regions of the world. In the Svalbard region of Norway, glaciers had a surge phase of 4–10 years and a quiescent phase of 50–500 years [83]. The Storstrømmen Glacier in Greenland had a surge phase of 10 years [84], while the Otto Fiord Glacier on Ellesmere Island in the Arctic had a surge cycle of about 50 years [85]. Glaciers in Iceland generally had a surge phase of two to three years, with surge cycles ranging from a few years to several hundred years [86]. Donjek Glacier in Canada had a surge phase of two years and a surge cycle of twelve years [87]. The Variegated Glaciers in Alaska demonstrated a surge phase of 1–2 years and a quiescent phase of 12–18 years [12]. Turner Glacier experiences surges at five-year intervals [88]. Currently, studies on the disparity of glacier surge cycles between HMA and other global regions are scarce, and this information is crucial for accurately predicting the timing of glacier surge in different regions and formulating effective measures for disaster prevention. It necessitates precise records of glacier surge events and puts forward more urgent requirements for the establishment of a comprehensive database of global surging glaciers.

*5.3. Possible Mechanism of Glacier Surging in HMA*

At present, the instability of glacial dynamics is generally believed to be the fundamental cause of glacial surges [89–91]. The control mechanisms for surging glaciers mainly include thermal and hydrological explanations [86,92–94]. Some scholars have employed a coupling effect based on mass and enthalpy budgets to comprehend glacier dynamics behaviour [22,95]. Due to the extensive and complex topography as well as climatic diversity of HMA, in situ observations of surging glaciers are challenging in this region. Therefore, researchers have investigated the mechanisms of glacier surging by comparing their characteristics with existing explanations in different regions of HMA. Studies generally agree that the surging of glaciers in the Pamir is primarily controlled by the thermal mechanism, wherein a continuous increase in mass within the reservoir zone leads to the bottom of the glacier reaching its pressure melting point and triggering a surge [14,33,69,73]. Existing studies present divergent perspectives on the mechanisms behind surging glaciers in the Karakoram Range. Some studies suggest that thermal processes dominate as triggers for Karakoram glacier surges [28,96], while others propose changes in hydrological conditions as the primary cause [8,79]. Additionally, some studies suggest a combination of both hydrological and thermal mechanisms [91]. Yasuda and Furuya [36] argue that a combined effect of thermal and hydrological mechanisms causes glacier surges in the West Kunlun region. The control mechanisms of the Northern Inylchek and Samoilowich glaciers in the Tian Shan differ [40,97]. The surging characteristics of glaciers in Geladandong Peak of Tanggula Shan are not entirely explained by hydrological and thermal mechanisms, as they may be influenced by multiple factors [41]. Previous studies only examined surging glacier features based on remote sensing images without field investigations into internal temperature and subglacial water systems. Therefore, further research is needed to understand the mechanism behind surging glaciers in HMA.

## 6. Conclusions

In this study, we identified surging glaciers in HMA based on Landsat TM/ETM+/OLI images acquired from 1986 to 2021 and some thematic datasets including SRTM/ASTER DEMs and ITS_LIVE glacier velocity. The rapid advance of a surging glacier terminus can pose a threat to downstream areas, potentially causing disasters. Therefore, this study focused on quantifying glaciers with significant advancements at their terminus. A total of 244 surging glaciers in HMA were identified, with the majority being medium- to large-sized glaciers with an area larger than 100 km$^2$. Existing research and our study results both indicate that surging glaciers exhibit a clustered distribution in space, with the Karakoram Range and Pamirs being the regions of the highest concentration of surging glaciers (185 glaciers, 75.8%).

With the support of long-term time series remote sensing images, we quantified the advance frequency of each glacier. This will contribute to a better understanding of the

surging process and characteristics of glaciers. The results show that from 1986 to 2021, a total of 2802 advance events occurred in 244 surging glaciers in HMA, exhibiting different spatiotemporal trends. Among different mountain regions in HMA, the highest number of surging glacier advances occurred in the Karakoram Mountains (1401), followed by the Pamirs (902) and Kunlun Mountains (261). The annual surge times occurring in 1993–2016 were all greater than 80. However, this phenomenon gradually decreased after the year 2016.

We identified a total of 36 glaciers in HMA that experienced two or more glacier surges from 1986 to 2021. Among them, the highest number was observed in the Pamirs (19), followed by the Karakorum (13), with fewer occurrences in other regions. This provides important evidence for understanding the periodicity of surging glaciers in HMA.

The dynamics of glacier surges, including their surge phase, quiescent phase, and surge cycle, exhibit regional variations within HMA. In the Karakoram Range and Pamir regions specifically, glacier surges tend to be relatively short, typically lasting between 2 and 6 years, with a quiescent phase of 5~19 years and a surge cycle of 9~24 years. The mechanisms governing glacier surging in HMA are intricate and likely influenced by multiple factors. This study is expected to provide fundamental data for further research on surging glaciers.

**Supplementary Materials:** The following supporting information can be downloaded at: https://www.mdpi.com/article/10.3390/rs15184595/s1.

**Author Contributions:** Conceptualization, X.Y.; methodology, S.Z. and X.Y.; validation, H.D. and Y.Z.; data curation, S.Z.; writing—original draft preparation, S.Z. and M.S.; writing—review and editing, X.Y., M.S., H.D. and Y.Z.; visualization, S.Z., H.D. and Y.Z.; supervision, X.Y.; funding acquisition, X.Y. All authors have read and agreed to the published version of the manuscript.

**Funding:** This work was supported by the National Natural Science Foundation of China (grant nos. 42161027 and 42071089), Strategic Action Plan of Oasis Science (grant no. NWNU-LZKX-202301), the Third Xinjiang Scientific Expedition Program (grant no. 2021xjkk0801), the Open Research Fund of the National Earth Observation Data Center (grant no. NODAOP2020007), and Gansu Province Education Science and Technology Innovation Project (grant no. 2021QB-019).

**Data Availability Statement:** The dataset generated for this study is provided in the Supplementary Materials: "surging glacier_HMA.zip".

**Conflicts of Interest:** The authors declare no conflict of interest. The funders had no role in the design of this study; in the collection, analyses, or interpretation of data; in the writing of the manuscript; or in the decision to publish the results.

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
