# Peer review of "Surging Glaciers in High Mountain Asia between 1986 and 2021"

_remotesensing, doi:10.3390/rs15184595_

Round 1

Reviewer 1 Report

Based on the Landsat image series from 1986-2021, the authors systematically studied the surging glaciers in the HMA region, including the total number, the total area, the main distribution subregions, the total time of surges, as well as surge phase and the quiescent phase of glaciers, and the surge cycles. This article is of high scientific value in providing a scientific knowledge for clarifying the surging glaciers in the HMA area. But I have some questions and suggestions about the manuscript specifically as follows. I will recommend the article for publication after the author has revised and refined these issues.

Line 15: Adding "in this study" at the appropriate place in the sentence, such as after "in HMA", will make the sentence flow more logically.

Line 40: The sentence should be “A glacier surge is a rapid movement of a glacier over a short period [11].”. Please note how the definite article is used.

Line 63: The word "have" in the sentence "Vale et al. [23] have identified" should be deleted.

What is the significance of this study? The authors should explain it in the introduction.

Line 97, In the legend of Figure 1, "RGI Regions" has been changed to "RGI Region".

Line101: Please indicate when the site was last accessed.

Line 103: This should be 3. Materials and Methods not 2. Materials and Methods.

Line 104: As I know, Second Chinese Glacier Inventory is included in RGI V6.0, so why does the author still use Second Chinese Glacier Inventory here?

Line108-109: Similarly, please indicate the date on which these sites were last accessed. Provide the time of the last visit for all URLs in the full text.

10 Line 112-113: “This extensive dataset offers substantial support for analyzing long-term glacier changes.” Please give examples.

11 Line 122-124: “Notably, the number of available images remained relatively low until 1990; however, it consistently exceeded 200 scenes per year thereafter, with only a few exceptions in certain years (1991, 1992, 1995, 1997, and 2021) (Figure 2b).” How did the authors overcome the impact on surging glacier identification because of the increase in imagery data after 1990?

12 Line 169: What does "visually revised" mean? I think it means "manually revised".

13 Line 234-235: “However, surging glaciers with a terminal advance of less than five pixels were excluded due to their classification as advancing glaciers [40,75].” Since Landsat's different sensors have different spatial resolutions, should the authors indicate here what image the 5 image elements are for? Or it might be better to use specific forward distances (with the number of image elements corresponding to different images in parentheses). This issue is important for the identification of surging glaciers.

14 Line 254: “we have identified”: “have” should be removed.

Minor editing of English language required.

Author Response

Based on the Landsat image series from 1986-2021, the authors systematically studied the surging glaciers in the HMA region, including the total number, the total area, the main distribution subregions, the total time of surges, as well as surge phase and the quiescent phase of glaciers, and the surge cycles. This article is of high scientific value in providing a scientific knowledge for clarifying the surging glaciers in the HMA area. But I have some questions and suggestions about the manuscript specifically as follows. I will recommend the article for publication after the author has revised and refined these issues.

Thanks for your approval of this study.

1 Line 15: Adding "in this study" at the appropriate place in the sentence, such as after "in HMA", will make the sentence flow more logically.

Thanks for your suggestions, and we added “in this study” in the following sentence.

Based on Landsat TM/ETM+/OLI remote sensing images from 1986-2021, a total of 244 surging glaciers were identified in HMA in this study, covering an area of 11724 km2 and accounting for 12.01% of the total area of glaciers in this region.

2 Line 40: The sentence should be “A glacier surge is a rapid movement of a glacier over a short period [11].”. Please note how the definite article is used.

Thanks for your suggestions. The sentence was modified as follows:

A glacier surge is a rapid movement of a glacier over a short period [11].

3 Line 63: The word "have" in the sentence "Vale et al. [23] have identified" should be deleted.

Thanks for your suggestions. The sentence was modified as follows:

Recently, Vale et al. [23] identified a total of 137 surging glaciers in HMA between 1987-2019 utilizing the GEE platform and the GEEDiT tool.

4 What is the significance of this study? The authors should explain it in the introduction.

Thanks for your suggestions, and we have made the following changes based on the comments.

The objective of this study is: (1) to identify surging glaciers in HMA between 1986-2021 from long time series of Landsat images; (2) to determine the occurrence time and frequency of these surging glaciers based on available yearly Landsat images; and (3) to investigate the periodicity and mechanisms of surging glaciers in HMA. It can provide a scientific basis and data support for understanding the characteristics of glacier changes in HMA and for glacier disaster prevention and mitigation efforts.

5 Line 97, In the legend of Figure 1, "RGI Regions" has been changed to "RGI Region".

Thanks for your suggestions. The modified Figure 1 was as follows:

6 Line101: Please indicate when the site was last accessed.

Thanks for your suggestions and we added the date information.

They can be downloaded from the National Tibetan Plateau Data Center (https://data.tpdc.ac.cn/, accessed on 20 October 2021) and the GLIMS website (https://www.glims.org/, accessed on 2 December 2021), respectively.

7 Line 103: This should be 3. Materials and Methods not 2. Materials and Methods.

I am sorry for my carelessness. The number of this section was modified as 3.

  1. Materials and Methods

8 Line 104: As I know, Second Chinese Glacier Inventory is included in RGI V6.0, so why does the author still use Second Chinese Glacier Inventory here?

Thanks for your suggestions. Most glaciers in China in RGI V6.0 adopt the Second Chinese Glacier Inventory (SCGI), which is the most authoritative glacier dataset in China. In the Karakoram region, unpublished glacier data completed by the Technical University of Dresden and the University of Zurich were employed in RGI V6.0. Unfortunately, the vectorized boundaries of glaciers in the Karakoram Range were extracted by a machine classification method and were not manually revised, which caused some snow patches to be included. Therefore, we adopted the SCGI to replace some glaciers that intersected with those in the former dataset.

9 Line108-109: Similarly, please indicate the date on which these sites were last accessed. Provide the time of the last visit for all URLs in the full text.

Thanks for your suggestions and we added the date information.

The background image is Natural Earth II with Shaded Relief, Water, and Drainages data from the Natural Earth website (https://www.naturalearthdata.com/, accessed on 22 March 2022).

which were obtained from the United States Geological Survey (USGS) website (https://earthexplorer.usgs.gov, accessed on 25 January 2022).

The above DEM data were acquired from the NASA EARTH DATA website (https://earthdata.nasa.gov/, accessed on 13 March 2022).

downloaded from NASA’s Jet Propulsion Laboratory, California Institute of Technology website (https://its-live.jpl.nasa.gov/, accessed on 29 January 2022).

10 Line 112-113: “This extensive dataset offers substantial support for analyzing long-term glacier changes.” Please give examples.

Thanks for your suggestions. We have added the relevant literature in this section.

The Landsat satellite series is renowned for providing the longest and most com-prehensive archive of Earth observation data [51,52]. This extensive dataset offers substantial support for analyzing long-term glacier changes [4, 14, 23, 34, 40-41].

11 Line 122-124: “Notably, the number of available images remained relatively low until 1990; however, it consistently exceeded 200 scenes per year thereafter, with only a few exceptions in certain years (1991, 1992, 1995, 1997, and 2021) (Figure 2b).” How did the authors overcome the impact on surging glacier identification because of the increase in imagery data after 1990?

Thanks for your suggestions. The increase in the number of images is beneficial for the identification of surging glaciers, as it provides a greater availability of imagery to support the identification of changes occurring at the terminus and the frequency of surges. The increase in quantity, however, poses greater challenges to image acquisition and processing. Therefore, we employed Python for batch processing. Moreover, the growing number of images further contributes to the uncertainty of surging glacier identification. In this study, we conducted an analysis on the errors encountered during glacier boundary and length extraction.

12 Line 169: What does "visually revised" mean? I think it means "manually revised".

We agree to your proposal and the word was modified.

Moreover, glacier outlines were manually revised with the aid of the SCGI and RGI V6.0 datasets.

13 Line 234-235: “However, surging glaciers with a terminal advance of less than five pixels were excluded due to their classification as advancing glaciers [40,75].” Since Landsat's different sensors have different spatial resolutions, should the authors indicate here what image the 5 image elements are for? Or it might be better to use specific forward distances (with the number of image elements corresponding to different images in parentheses). This issue is important for the identification of surging glaciers.

Thanks for your suggestions. We added the quantified description in the modified sentence.

However, surging glaciers with a terminal advance of less than five pixels (TM) and ten pixels (ETM+ and OLI) were excluded due to their classification as advancing glaciers [40,75].

14 Line 254: “we have identified”: “have” should be removed.

Thanks for your suggestions and we deleted the word “have”.

Based on the SCGI and RGI V6.0 datasets, in conjunction with Landsat TM/ETM+/OLI remote sensing images from 1986-2021, we identified a total of 244 surging glaciers in HMA by conducting comparative analyses of terminal positions, glacier surface features, glacier surface elevation changes, glacier flow velocity changes, and terminal shapes.

Reviewer 2 Report

In this manuscript the authors investigate and analyze the surging glaciers of the HMA based on data such as Landsat optical images from 1986 to 2021. The authors identify 244 surging glaciers and explore the spatial and temporal trends they present, analyzing the surge cycles of 36 glaciers and their regional variations. One of the most valuable parts of this study is the analysis of the bimonthly terminus changes over a 35-year period throughout the region. Overall, except for a few works that need to be clarified, this article is well written.

Specific Comments:

L129:How to produce the pan-sharpened images and provide the methods used.

L166-167:How to assess and validate errors due to manual corrections and how to ensure consistency during manual identification?

L171-174-189:How to consider the error caused by image registration? Why only consider errors due to spatial resolution?

 L209-210:Equation 3-4 and its description are not clear.

L216:What is the basis for choosing 600m?

L229:What is the spatial resolution corresponding to five pixels?

L234-238:The second criterion for surging glacier identification is subjective and arbitrary and may exclude potential surging glaciers, and therefore requires further clarification, otherwise it will affect the accuracy of the results.

L246-427:Equation 1-4 in the methods section 3.2 describe the uncertainty how to calculate, why is there no error analysis in Section 4?

L313 and L348:What is the basis for choosing these two glaciers as typical examples?

L348:Incorrect section numbers.

L337-338 and L377-378:what is accumulation area, receiving area; accumulation zone and how to delineate these zones?

L414:“with the majority falling between 414 9 to 22 years, How about others?

Author Response

In this manuscript the authors investigate and analyze the surging glaciers of the HMA based on data such as Landsat optical images from 1986 to 2021. The authors identify 244 surging glaciers and explore the spatial and temporal trends they present, analyzing the surge cycles of 36 glaciers and their regional variations. One of the most valuable parts of this study is the analysis of the bimonthly terminus changes over a 35-year period throughout the region. Overall, except for a few works that need to be clarified, this article is well written.

Thanks for your approval of our manuscript.

L129: How to produce the pan-sharpened images and provide the methods used.

Thanks for your suggestions. We added the method description in the modified sentences.

Subsequently, colour composites were generated by combining visible bands of Landsat TM/ETM+/OLI (Landsat TM and ETM+: Bands 5, 4, 3; Landsat OLI: Bands 6, 5, 2) using the CompositeBands tool in ArcGIS 10.4 software. Moreover, pan-sharpened images with a spatial resolution of 15 m for Landsat ETM+/OLI were produced by fusing the colour composite with the panchromatic band using the CreatePansharpenedRasterDataset tool in ArcGIS 10.4 software. The above processes were batch processed using Python code to enhance efficiency.

L166-167: How to assess and validate errors due to manual corrections and how to ensure consistency during manual identification?

Thanks for your suggestions. All images provided by USGS underwent radiometric and geometric correction, as well as topographic correction based on DEM data. In the process of surging glacier identification, we visually checked the positional relationship of the contrasting images in different periods, which approved that most images had good spatial matching. For manual identification, most participants participated in the Second Chinese Glacier Inventory and have rich experiences in glacier interpretation, which ensured that the glacier boundary crossed through the middle of mixed pixels. Therefore, we use Equation (1) to calculate its error.

L171-174-189: How to consider the error caused by image registration? Why only consider errors due to spatial resolution?

Thanks for your suggestions. Due to the difficulties in quantifying the errors caused by satellite sensors and image registration, we only consider the error caused by image resolution. The modification is as follows:

The accuracy of glacier outline extraction is mainly affected by remote sensing sensors, image registration [62], and pixel offset errors resulting from subjective visual interpretation [57,58,63]. The errors caused by satellite sensors and image registration are difficult to quantify [41,61]. Here, we just consider the error resultant from spatial resolution of remote sensing images, which can be calculated by the following equation [40,63]:

L209-210: Equation 3-4 and its description are not clear.

Thanks for your suggestions, and we have made the following changes based on your comments.

Prior to calculating changes in glacier surface elevation, it is essential to co-register different DEM data due to variations in their acquisition and processing methods. Therefore, we used the method proposed by Nuth and Kääb [68] to co-register the ASTER DEM and SRTM DEM. Meanwhile, during the matching process, a threshold of ± 100 m was chosen to reject anomalies in elevation difference, and the area where the ground slope is less than 5° was excluded to improve matching accuracy [69]. The harsh environment in glacier regions makes it difficult to evaluate the error of DEM data by field measurements. Therefore, assuming that the elevation of non-glacier areas remains constant over time, the uncertainty of elevation change between multiple DEMs is assessed by calculating the mean elevation difference (MED) and standard deviation (SD) in non-glacier areas [70].

where e is the error of elevation change; STDVno glac is the standard deviation of the non-glacial area; and n is the number of pixels within the non-glacial area. Considering the strong spatial autocorrelation among neighboring pixels in DEM data, it is possible to neglect such correlation when the distance between pixels exceeds 20 times the spatial resolution of pixels [71]. Therefore, we adopted 600 m as the de-spatial autocorrelation distance.

L216: What is the basis for choosing 600m?

Thanks for your suggestions. The DEM data exhibits strong spatial autocorrelation between adjacent pixels. This autocorrelation can be ignored when the distance between pixels exceeds 20 times the spatial resolution of the pixels. In this study, the DEM used has a resolution of 30 m, so the distance chosen is 600 m.

L229: What is the spatial resolution corresponding to five pixels?

Thanks for your suggestions, and we added the corresponding information of spatial resolution for different Landsat sensors.

However, surging glaciers with a terminal advance of less than five pixels (TM) and ten pixels (ETM+, OLI) were excluded due to their classification as advancing glaciers [40,75].

L234-238: The second criterion for surging glacier identification is subjective and arbitrary and may exclude potential surging glaciers, and therefore requires further clarification, otherwise it will affect the accuracy of the results.

Thanks for your suggestions. In this study, the identification of surging glaciers is primarily based on the first criterion, with the second and third criteria serving as supplementary. We referred to the literature [24] for the second criterion.

L246-427: Equation 1-4 in the methods section 3.2 describe the uncertainty how to calculate, why is there no error analysis in Section 4?

Thanks for your suggestions. In section 4.1, for statistical purposes, the data used to calculate glacier area are from RGI and SCGI, not the revised boundary data. Therefore, no error calculation was performed on these data. However, in section 4.3, we calculated the errors for glacier area change and glacier elevation change, and appended the respective error values to the corresponding measurements.

L313 and L348: What is the basis for choosing these two glaciers as typical examples?

Thanks for your suggestions. The reason for choosing these two glaciers is that one exhibited the greatest terminus advancement during the surge phase among all surging glaciers, while the other possessed multiple surging tributary glaciers.

L348: Incorrect section numbers.

It’s my mistake. The number was added in the modified manuscript.

4.3.2. Musta Glacier

L337-338 and L377-378: what is accumulation area, receiving area; accumulation zone and how to delineate these zones?

Thanks for your suggestions. We have referred to the explanation and delineation proposed by Meier and Post regarding the accumulation zone and receiving area. The ice reservoir is not identical with the accumulation zone; the reservoir can be entirely within the ablation zone. During the quiescent phase the reservoir thickens and the longitudinal profile in the lower part of the reservoir area continuously steepens. The active phase usually appears to begin with rapid movement where this steepening has occurred. The rapid movement propagates quickly up glacier to include all the ice reservoir area and down glacier into the receiving area. The rapid flow removes ice from the reservoir area causing vertical lowering of the surface in the order of tens or hundreds of meters, and adds ice to the receiving area causing a similar or greater amount of thickening there.

L414: “with the majority falling between 414 9 to 22 years”, How about others?

I’m sorry for the false number “414”. The right sentence is “The surge cycle ranges from 5 to 30 years, with the majority falling within the 9 to 22 years.”

Reviewer 3 Report

Glacier surging is an important glacier movement phenomenon, in which the speed alternates between slow and fast and regular. It is usually not associated with climate factors but can have considerable consequences, such as ice collapse and avalanches, blocking river valleys, forming ice-dammed lakes, etc. So the investigation of surging glaciers is of importance and significance. This manuscript tried to form a dataset of glacier surges over the High Mountain Asia (HMA) from 1986 to 2021 using available Landsat TM/ETM+/OLI remote sensing images combined with SRTM DEM and ASTER GDEMs. The dataset of surging glaciers in HMA help to understand the complexity of glacier change in this region. It can be published in Remote Sensing, but some corrections are still made before publication. The main comments are as follows.

Main comments

1. As mentioned in your article, most scholars have conducted a comprehensive inventory of the surging glaciers in the HMA region and even all over the world. Please explain what is difference of this study from these studies. In particular, Guo et al. [1] have also provided a long-term series dataset of glacial surges in the HMA since the 1970s. It appears that this study is the most recent dataset on glacial surges in the HMA that has received recognition. However, it's important to understand the strengths and innovations of this study compared to this dataset and the other previous datasets. Can you please provide an explanation of how your study differs from this dataset, and what advancements have been made? Additionally, it would be helpful to include a comparison of the results between the two datasets.

2. Lines 496-499 mentioned that this study identified a relatively lower number of surging glaciers in HMA, so there is reason to suspect that the surging glacier identified in this study is inaccurate. The article needs to provide more information and evidence to support the identification results of the surging glaciers. Additionally, it should include detailed descriptions for better understanding. Please increase the verification of the methods and results for identifying the surging glaciers.

3. From the text, the glacier outlines are extracted using a combination of automatic computer classification and visual interpretation in this study. However, the article lacks a detailed description of the automatic computer classification method's implementation process and formula, please describe this method in detail.

4. Surging glaciers were identified based on the changes in glacier surface elevation calculated by the DEM data in this research (lines 234-245). However, the time scales of the DEM data used in this study (SRTM and ASTER DEM data) only include the period after 2000, how was glacial surging before 2000 identified in advance by changes in elevation?

5. In section 6, we observed that the authors present an analysis of surging glaciers in HMA from three perspectives: similarities and differences, periodicity, and mechanism. While we appreciate the author's discussion from these three aspects, there are still two issues to address. Firstly, it is customary for the mechanism section to be extensively examined in studies on surging glaciers. Therefore, it would be more appropriate for the authors to draw their own conclusions based on research findings rather than solely relying on existing literature references. Secondly, when discussing glacier changes, it is often essential to consider the context of climate change; however, this aspect seems to have been overlooked. I believe that incorporating a discussion on climate change enhances its comprehensiveness.

Reference:

1.      Guo, L.; Li, J.; Dehecq, A.; Li, Z.W.; Li, X.; Zhu, J.J. A new inventory of High Mountain Asia surging glaciers derived from multiple elevation datasets since the 1970s. Earth System Science Data 2023, 15, 2841–2861, https://doi.org/10.5194/essd-15-2841-2023.

Minor comments:

1. The title of the “Materials and Methods” is in the wrong order.

2. The lack of the section 4.3.2.

3. The author's labeling of B1 and B2, as shown in Figure 9 but not mentioned in the analysis.

4. It is necessary to further improve English.

5. The conclusion section only summarizes the article's results and fails to condense the core conclusions, which need further modification.

It is necessary to further improve English.

Author Response

Glacier surging is an important glacier movement phenomenon, in which the speed alternates between slow and fast and regular. It is usually not associated with climate factors but can have considerable consequences, such as ice collapse and avalanches, blocking river valleys, forming ice-dammed lakes, etc. So the investigation of surging glaciers is of importance and significance. This manuscript tried to form a dataset of glacier surges over the High Mountain Asia (HMA) from 1986 to 2021 using available Landsat TM/ETM+/OLI remote sensing images combined with SRTM DEM and ASTER GDEMs. The dataset of surging glaciers in HMA help to understand the complexity of glacier change in this region. It can be published in Remote Sensing, but some corrections are still made before publication. The main comments are as follows.

Thanks for your approval of this study.

Major comments:

  1. As mentioned in your article, most scholars have conducted a comprehensive inventory of the surging glaciers in the HMA region and even all over the world. Please explain what is difference of this study from these studies. In particular, Guo et al. [1] have also provided a long-term series dataset of glacial surges in the HMA since the 1970s. It appears that this study is the most recent dataset on glacial surges in the HMA that has received recognition. However, it's important to understand the strengths and innovations of this study compared to this dataset and the other previous datasets. Can you please provide an explanation of how your study differs from this dataset, and what advancements have been made? Additionally, it would be helpful to include a comparison of the results between the two datasets.

Thanks for your suggestions. Our research mainly focuses on glaciers that undergo significant terminus advances. Utilizing available remote sensing imagery, we conducted a statistical analysis of the frequency of terminus advances for each surging glacier. Moreover, our study included glaciers with two or more surges, and we also analyzed their periodicity. I believe this is the major distinction and advantage of our research. On the other hand, Guo et al.'s study provides us with a valuable inventory of surging glaciers spanning a longer time scale. Their research primarily relies on changes in glacier surface elevation as the identification criteria, with less emphasis on the advances at the glacier terminus.

  1. Lines 496-499 mentioned that this study identified a relatively lower number of surging glaciers in HMA, so there is reason to suspect that the surging glacier identified in this study is inaccurate. The article needs to provide more information and evidence to support the identification results of the surging glaciers. Additionally, it should include detailed descriptions for better understanding. Please increase the verification of the methods and results for identifying the surging glaciers.

Thanks for your suggestions. Our research mainly focuses on glaciers that underwent significant terminus advances. Glaciers with terminus advances of less than 150 meters have been excluded, which is the reason for the relatively low number of identified surging glaciers in this study. We will provide the data of surging glaciers identified in this research, including attribute information such as the start and end times of surges, and information on changes in glacier surface elevation.

  1. From the text, the glacier outlines are extracted using a combination of automatic computer classification and visual interpretation in this study. However, the article lacks a detailed description of the automatic computer classification method's implementation process and formula, please describe this method in detail.

Thanks for your suggestions. In previous studies, we used the same method to extract glacier outlines. For detailed procedures, please refer to the paper by Zhou et al., 2021.

Zhou, S.; Yao, X.; Zhang, D.; Zhang, Y.; Liu, S.; Min, Y. Remote Sensing Monitoring of Advancing and Surging Glaciers in the Tien Shan, 1990–2019. Remote Sens. 2021, 13. doi:10.3390/rs13101973.

  1. Surging glaciers were identified based on the changes in glacier surface elevation calculated by the DEM data in this research (lines 234-245). However, the time scales of the DEM data used in this study (SRTM and ASTER DEM data) only include the period after 2000, how was glacial surging before 2000 identified in advance by changes in elevation?

Thanks for your suggestions. Due to the difficulty in obtaining DEM data prior to 2000, we only extracted glacier surface elevation changes for a subset of glaciers after the year 2000. This is clarified in the data source acquisition section.

  1. In section 6, we observed that the authors present an analysis of surging glaciers in HMA from three perspectives: similarities and differences, periodicity, and mechanism. While we appreciate the author's discussion from these three aspects, there are still two issues to address. Firstly, it is customary for the mechanism section to be extensively examined in studies on surging glaciers. Therefore, it would be more appropriate for the authors to draw their own conclusions based on research findings rather than solely relying on existing literature references. Secondly, when discussing glacier changes, it is often essential to consider the context of climate change; however, this aspect seems to have been overlooked. I believe that incorporating a discussion on climate change enhances its comprehensiveness.

Thanks for your suggestions. Clarifying the mechanism of glacier surges requires more field measurements, especially regarding information about the glacier bed and glacier temperatures, which are extremely challenging to obtain. Therefore, relying solely on remote sensing methods makes it difficult to provide a clear understanding of the mechanisms behind surging glaciers. In the initial version, we attempted to draw our own conclusions based on our results, but faced skepticism from multiple reviewers. As a result, we only relied on existing literature to provide a brief discussion. The impact of climate change on glacier changes cannot be ignored. However, for surging glaciers, many studies have shown that their occurrence is not solely influenced by climate change. Similarly, this aspect was questioned by reviewers in the initial manuscript, which is why we did not discuss the climate background in the paper. We hope you can understand.

Minor comments:

  1. The title of the “Materials and Methods” is in the wrong order.

I am sorry for my carelessness. The number of this section was modified as 3.

  1. Materials and Methods
  2. The lack of the section 4.3.2.

It’s my mistake. The number was added in the modified manuscript.

4.3.2. Musta Glacier

  1. The author's labeling of B1 and B2, as shown in Figure 9 but not mentioned in the analysis.

Thanks for your suggestions. We removed the labels of B1 and B2 in Figure 9.

  1. It is necessary to further improve English.

Thanks for your suggestions. We have made some modifications to the language.

  1. The conclusion section only summarizes the article's results and fails to condense the core conclusions, which need further modification.

Thanks for your suggestions, and we have made the following changes based on the comments.

In this study, we have identified surging glaciers in HMA based on Landsat TM/ETM+/OLI images acquired from 1986 to 2021 and some thematic datasets including SRTM/ASTER DEMs and ITS_LIVE glacier velocity. The rapid advance of surging glacier terminus can pose a threat to downstream areas, potentially causing disasters. Therefore, this study focuses on quantifying glaciers with significant advancements at their terminus. A total of 244 surging glaciers in High Asia were identified, with a majority being medium to large-sized glaciers with an area larger than 100 km2. Existing research and our study results both indicate that surging glaciers exhibit a clustered distribution in space, with the Karakoram Range and Pamirs being the regions of highest concentration for surging glaciers (185 glaciers, 75.8%).

With the support of long-term time series remote sensing images, we quantified the advance frequency of each glacier. This will contribute to a better understanding of the surging process and characteristics of glaciers. The results show that from 1986 to 2021, a total of 2,802 advance events have occurred in 244 surging glaciers in HMA, exhibiting different spatiotemporal trends. Among different mountain regions in HMA, the highest number of surging glacier advances occurred in the Karakoram Mountains (1401), followed by the Pamirs (902) and Kunlun Mountains (261). The annual surge times occurring in 1993-2016 were all greater than 80. However, this phenomenon gradually decreased after the year of 2016.

We identified that a total of 36 glaciers in HMA experienced two or more glacier surges from 1986 to 2021. Among them, the highest number observed in the Pamirs (19), followed by the Karakorum (13) and fewer occurrences in other regions. This provides important evidence for understanding the periodicity of surging glaciers in HMA.

The dynamics of glacier surges, including their surge phase, quiescent phase, and surge cycle, exhibit regional variations within HMA. In the Karakoram Range and Pamir regions specifically, glacier surges tend to be relatively short, typically lasting between 2 to 6 years, with the quiescent phase of 5~19 years and the surge cycle of 9~24 years. The mechanisms governing glacier surging in HMA are intricate and likely influenced by multiple factors. This study is expected to provide fundamental data for further research on surging glaciers.

Round 2

Reviewer 2 Report

The author has responded to the questions in detail and revised accordingly, and the revision has no major problems and is agreed for publication.